# BMMR: A Large-Scale Bilingual Multimodal Multi-Discipline Reasoning Dataset

**Zhiheng Xi**[1][*][†] **Guanyu Li**[1][*] **Yutao Fan**[2,3][*] **Honglin Guo**[1][*]

**Yufang Liu**[4] **Xiaoran Fan**[1] **Jiaqi Liu**[1] **Jingchao Ding**[7] **Wangmeng Zuo**[3]

**Zhenfei Yin**[5,6][†]  **Lei Bai**[2] **Tao Ji**[1] **Tao Gui**[1,8][†] **Qi Zhang**[1] **Xuanjing Huang**[1]

[1]**Fudan University**  [2]**Shanghai AI Laboratory** [3]**Harbin Institute of Technology**
[4]**East China Normal University** [5]**Oxford** [6]**University of Sydney**
[7]**Yimudata**  [8]**Pengcheng Laboratory**

## Abstract

In this paper, we introduce BMMR, a large-scale bilingual, multimodal, multi-disciplinary reasoning dataset for the community to develop and evaluate large multimodal models (LMMs). BMMR comprises 110k college-level questions spanning 300 UNESCO-defined subjects, spanning diverse formats—multiple-choice, fill-in-the-blank, and open-ended QA—and sourced from both print and digital media such as books, exams, and quizzes. All data are curated and filtered via a human-in-the-loop and scalable framework, and each instance is paired with a high-quality reasoning path. The dataset is organized into two parts: BMMR-Eval that comprises $20,458$ high-quality instances to comprehensively assess LMMs' knowledge and reasoning across multiple disciplines in both Chinese and English; and BMMR-Train that contains $88,991$ instances to support further research and development, extending the current focus on mathematical reasoning to diverse disciplines and domains. In addition, we propose the process-based multi-discipline verifier (i.e., `BMMR-Verifier`) for accurate and fine-grained evaluation of reasoning paths. Extensive experiments on 24 models reveal that (i) even SOTA models (e.g., `o3` and `Gemini-2.5-Pro`) leave substantial headroom on BMMR-Eval; (ii) reasoning models exhibit discipline bias and outperform LMMs only on specific subjects; (iii) open-source models still trail their proprietary counterparts; and (iv) fine-tuning on BMMR-Train narrows this gap. Additionally, we conduct reasoning-chain analyses using `BMMR-Verifier` and other in-depth studies, uncovering the challenges LMMs currently face in multidisciplinary reasoning. We will release the data, and we hope our work can offers insights and contributions to the community.

Project Site: https://bmmr.pages.dev/
Code & Sources: https://github.com/WooooDyy/BMMR/

## 1 Introduction

Large multimodal models (LMMs) [1–3] and large reasoning models (LRMs) [4] have demonstrated extraordinary expertise and reasoning capabilities across a wide range of academic fields—such as mathematics, physics, and chemistry [5–7]. These models, represented by `GPT-4o` [8] and `OpenAI-o1` [9], can process and reason over both textual and visual inputs, and have generated

---

[*] Equal Contribution.

[†] Correspondence to: `zhxi22@m.fudan.edu.cn`, `tgui@fudan.edu.cn`, `zhenfei.yin@sydney.edu.au`

39th Conference on Neural Information Processing Systems (NeurIPS 2025) Track on Datasets and Benchmarks.

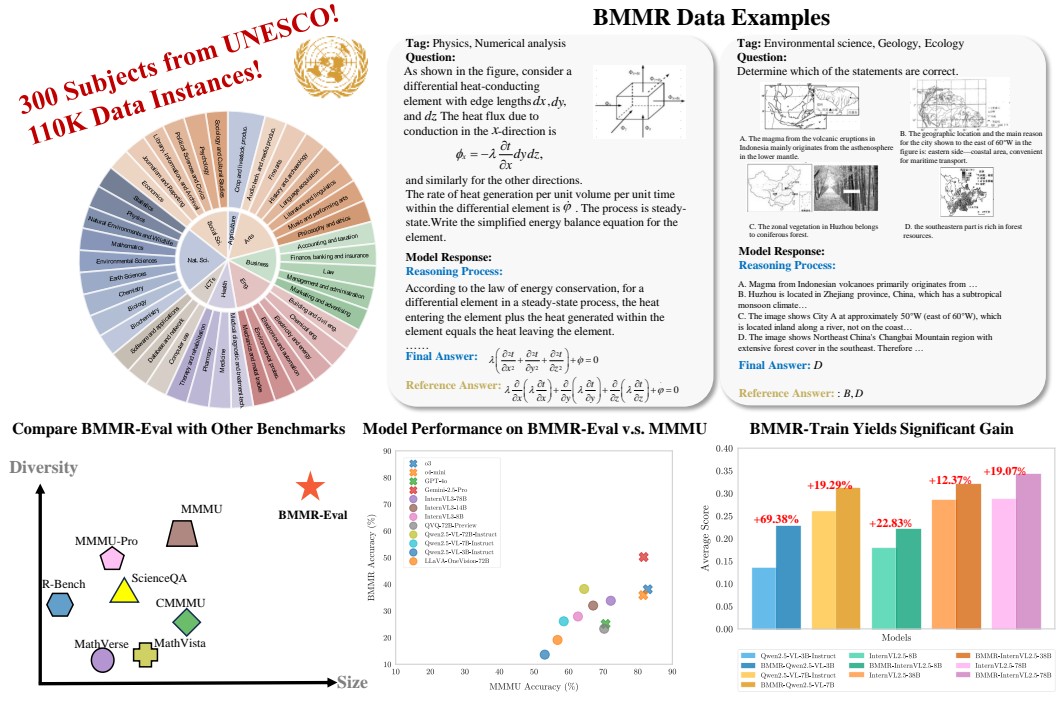

Figure 1: Overview of the BMMR dataset. It encompasses 110k instances across 300 subjects defined by UNESCO. We present two illustrative examples for visualization (top-middle and top-right). Furthermore, we compare our BMMR-Eval with other benchmarks regarding size and diversity (bottom-left). A comparison of model performance on BMMR-Eval versus MMMU is also included (bottom-middle), highlighting the challenging nature of our test set. Finally, we demonstrate that fine-tuning open-source models of various sizes (3B-78B) on our BMMR-Train yields significant performance enhancements (bottom-right).

significant interest in the AI community due to their potential to enable more general AI systems, i.e., AGI [10, 11].

However, with these advancements, comprehensively and accurately evaluating knowledge and reasoning capabilities of LMMs and LRMs across disciplines has become increasingly challenging. Existing benchmarks [12–14] struggle to strike a balance among subject diversity, problem complexity, reasoning depth, and language coverage, and have recently begun to exhibit performance saturation [15–18]. At the same time, the community lacks a multimodal, multidisciplinary training dataset—one that offers diverse questions and curated reasoning paths—to support research and development, especially within the open-source community [6, 19].

To bridge this gap, we introduce BMMR (Section 3): a large-scale bilingual, multimodal, multi-disciplinary reasoning dataset that contains 110k college-level high-quality instances, spanning 8 high-level disciplines and 300 sub-fields from UNESCO (The United Nations Educational, Scientific and Cultural Organization) [20], as illustrated in Figure 1. BMMR is organized into two parts: (1) BMMR-Eval, which comprises 20k instances with broad subject coverage and multiple difficulty levels for comprehensively assessing models' knowledge and reasoning across disciplines in both English and Chinese (see Table 1); and (2) BMMR-Train, which contains 89k instances to support further research and development, and extend the community's focus on mathematical reasoning to more diverse disciplines and domains.

We collect BMMR data from both digital and print sources—including books, exams, and quiz collections—and the dataset encompasses diverse formats such as multiple-choice, fill-in-the-blank, and open-ended QA. All instances are curated and filtered through a human-in-the-loop and scalable processing framework and paired with a high-quality reasoning path to ensure robustness and solidness. Every retained question in BMMR demands precise cross-modal comprehension, specialized domain knowledge, and advanced reasoning skills to solve [21–23].

To further enable accurate and fine-grained evaluation of models' reasoning abilities across disciplines and to prevent models from simply recalling or guessing the correct answers [24–26], we also propose `BMMR-Verifier`—a process-based bilingual, multimodal, multidisciplinary verifier (Section 4).

Extensive experiments on 24 LMMs and LRMs (Section 5.2 and Section 5.3) reveal that: (1) Even SOTA models perform suboptimally—for instance, `o3` and `Gemini2.5-Pro` only achieves 38.06 and 50.15, revealing substantial headroom; (2) Contrary to intuition, LRMs do not consistently outperform LMMs across all disciplines. Instead, they exhibit clear subject bias, excelling only in specific areas such as mathematical reasoning. This further validates BMMR's emphasis on multi-discipline knowledge; (3) Open-source models still lag behind their proprietary counterparts, highlighting the academia-industry gap. (4) Fine-tuning on BMMR-Train narrows this gap—for example, the finetuned `BMMR-InternVL2.5-78B` achieves a 19.07% improvement in overall performance.

Additionally, using the developed `BMMR-Verifier`, we conduct a fine-grained analysis of reasoning processes (Section 5.4). We present the distribution of reasoning-step quality across different models and examine, at a granular level, their reasoning abilities in various disciplines. Furthermore, through error categorization, qualitative studies, and deeper analyses (Section 6), we highlight key challenges in multimodal reasoning—such as overthinking [27, 28] and hallucination [29, 30]—and hope these findings offer valuable insights for advancing the next-generation models.

In summary, our main contributions are:

1. We introduce BMMR, a large-scale bilingual, multimodal, multidisciplinary reasoning dataset—comprising BMMR-Eval and BMMR-Train—to enable comprehensive evaluation and support research and development of multimodal foundation models.

2. We propose the multimodal, multidisciplinary, process-based `BMMR-Verifier` for accurate and fine-grained evaluation of the models' reasoning capabilities.

3. We conduct extensive experiments and analysis on 24 open-source and proprietary LMMs and LRMs, and provide key findings and insights. We hope our work can contribute to the field and inspire future research.

## 2 Related Work

**Benchmarks for LMMs.** The evaluation of multimodal models' intelligence remains a critical endeavor [31]. While fundamental benchmarks have been introduced to evaluate core visual understanding skills of LMMs, including visual classification [32], retrieval [33], grounding [34], and question-answering [35], they do not specifically focus on reasoning capabilities in multidisciplinary tasks. MMMU [5] notably pioneered multi-discipline understanding evaluation with its 11k problems spanning 30 subjects. However, such traditional multi-discipline benchmarks demonstrate insufficient logic reasoning demands, failing to challenge contemporary state-of-the-art LMMs such as `Gemini 2.5` [36] and `InternVL3` [2]. Recent research has shifted toward evaluating System-2 reasoning through advanced benchmarks requiring a significantly higher cognitive standard: MathVista [37] employs both multiple-choice and open-ended formats to probe mathematical reasoning, while MathVerse [38] systematically investigates modality-specific performance variations to isolate visual understanding impacts. Although these emerging benchmarks pose significant challenges for current LMMs [12, 39, 40], they still exhibit critical limitations in providing holistic assessments of reasoning abilities across multiple disciplines. In this work, we build the larger-scale BMMR-Eval that covers more diverse subjects (see Table 1).

**Multimodal reasoning datasets.** To advance the reasoning capabilities of LMMs, researchers have developed specialized multimodal training datasets [6, 43]. Current efforts include datasets targeting foundational visual reasoning tasks such as commonsense reasoning, embodied planning [44], and spatial reasoning [45–47]. For complex reasoning challenges, studies like LLaVA-CoT [48] and MAmmoTH-VL [49] generate structured reasoning paths across diverse visual reasoning domains, while ScienceQA [6] and MM-Eureka [50] offer multidisciplinary question-answer datasets with detailed chain-of-thought annotations. However, these resources remain constrained by their exclusive focus on K-12-level content, which limits their effectiveness in advancing state-of-the-art models that require higher-order reasoning. In this work, we address these limitations by constructing a new college-level multimodal dataset featuring cross-modal comprehension, specialized domain knowledge and advanced reasoning.

Table 1: Overall comparison between BMMR-Eval and other existing benchmarks. In the Source column, D means digital-based data sources, such as websites and existing datasets; P means print-based data sources, such as college textbooks and exams; R means repurposed data sources. The column Multiple Images implies the presence of questions that contains multiple images. In the Question Type column, MC means multiple-choice questions, FIB means fill-in-the-blank questions, ans OE means open-ended questions, TF means true-or-false questions. (t) in the Language column means "translated". In the Difficulty column, C means college level, K means K-12 level, and H means high-school level. Information for R-Bench only cover its multimodal subset. For all datasets, we only report statistics on their test split.

| | Source | #Item | #Discipline | Multiple Images | Reasoning Path | Question Type | Language | Difficulty |
|---|---|---|---|---|---|---|---|---|
| MMMU [5] | D, P | 10.5k | 6/30/183 | Yes | Partial | MC, OE | EN. | C |
| MMMU-Pro [35] | D, P | 1.7k | 6/30/183 | Yes | Partial | MC, OE | EN | C |
| CMMMU [41] | D, P | 11k | 6/30 | Yes | No | MC, FIB, TF | ZH | C |
| MathVista [37] | D | 6.1k | Math | No | Partial | MC, OE | EN | K, C |
| MathVerse [38] | D | 3.9k | Math | No | Partial | MC, OE | EN | H, C |
| ScienceQA [6] | P | 4.2k | 3/26/127 | No | Yes | MC | EN | K |
| R-Bench [42] | P | 665 | 83 | No | No | MC, TF | EN, ZH (t) | C |
| **BMMR-Eval (Ours)** | D, P, R | 20k | 8/16/40/264 | Yes | Yes | MC, FIB, OE | EN, ZH | C |

**Process reward models and verifiers.** Apart from final answer validation, process evaluation is also important for reasoning tasks [51, 52]. Research in LLMs has progressed from foundational Outcome-supervised Reward Models (ORMs) [24, 53, 54] that evaluate final outputs to more Process Reward Models (PRMs) [55, 25] designed to supervise intermediate steps in complex reasoning tasks. While PRMs, trained via methods including human annotation [51, 56] and Monte Carlo (MC) estimation [57–59, 25, 60, 61], offer finer-grained guidance, they suffer from inaccuracies, such as those arising from MC estimation bias and vulnerability to reward hacking. To address these limitations, verifiers have been introduced as a corrective mechanism [62–64], employing objective criteria like reference answers and formal rules to ensure the reliability of outputs and reasoning steps. In this work, we develop BMMR-Verifier to enhance the evaluation of models' reasoning paths across different disciplines, enabling a more granular assessment of their performance.

## 3 BMMR: A Bilingual Multimodal Multi-Discipline Reasoning Dataset

### 3.1 Overview of BMMR

The BMMR dataset is proposed to support the evaluation and development of multimodal foundation models in college-level, multidisciplinary knowledge, understanding, and reasoning. It comprises 110k items spanning 300 UNESCO-defined subfields across 8 high-level disciplines.

BMMR is bilingual (English and Chinese) and sourced from both print and digital media, including books, exams, and quizzes. This variety of sources inevitably introduces uncertainty in data quality. We design specific procedures to ensure question diversity, complexity, and answer verifiability. We also re-organize the original questions—through rewriting and augmentation—into multiple-choice, fill-in-the-blank, and open-ended QA formats to minimize the impact of model memorization and guessing. Each retained instance requires cross-modal understanding, domain-specific expertise, and advanced reasoning skills to solve. To support the research community, each instance is paired with a high-quality reasoning path.

BMMR is splited into two subsets: BMMR-Eval, containing $20,458$ examples, and BMMR-Train, containing $88,991$ examples. Specifically, BMMR-Eval is designed to comprehensively assess LMMs' perception, knowledge, and reasoning across a broad range of disciplines and difficulty levels; BMMR-Train supports the community's research and development of next-generation multimodal foundation models, extending the current focus of the community on mathematical reasoning to diverse disciplines and domains. The statistics of BMMR is listed in Table 4 in Appendix B.

## 3.2 Data Collecting and Curation Framework for BMMR

By conducting multiple rounds of human-in-the-loop review and revision, we ultimately develop a solid and scalable data collection and curation framework comprising six main steps: (1) taxonomy gathering; (2) data collection and preprocessing; (3) discipline classification and tagging; (4) safety and objectivity checks and self-consistency validation; (5) data transformation and augmentation; and (6) quality control and distribution balancing. The full workflow is detailed in Appendix A.

## 4 BMMR-Verifier: A Process-based Multimodal, Multi-Discipline Verifier

**Motivation.** Rule-based answer extraction and exact-match scoring simplify the comparison between a model's output and the reference answer. However, this approach introduces several challenges: (1) false positives, where a model arrives at the correct answer through flawed reasoning [65, 66]; (2) memorization and guessing, where the model simply recalls the answer without performing meaningful reasoning [67–69]; and (3) misjudgments, where the model's answer is actually correct but fails to exactly match the reference annotation [70, 71].

As we aim to accurately evaluate the model's reasoning path at a fine-grained level—and to minimize misjudgments—we introduce BMMR-Verifier, a process-based, multidisciplinary multimodal verifier. Given a question, a reference solution, and a model response, BMMR-Verifier precisely scores each step of the model's reasoning path and determines the correctness of the final answer.

**Training receipe of BMMR-Verifier.** Given a dataset $\mathcal{D} = \{x, r\}$, where $x$ denotes the input (comprising both the images and the query) and $r$ represents the reference solution. We perform 32 rollouts per sample from multiple models. A correctness label $c$ is assigned to each trajectory $\tau$ via rule-based evaluation. As a result, we obtain an augmented dataset $\mathcal{D}_r = \{x, r, \tau, c\}_{i=1}^{N}$ consisting of $N$ tuples. We perform an additional rebalancing and filtering step to balance the difficulty distribution of the dataset and to filter out low-quality samples, resulting in a curated training set $\mathcal{D}_v$.

Next, we employ the same method in Wang et al. [25], Yu et al. [59] to assign step-level scores to each reasoning trajectory $\tau$. Given the ground-truth label $c$, we assign a positive "+" or negative "−" tag as the label $y$. We then insert the label $y$ to the end of every step and get the new trajectory

$$\tau^* = \{ s_1, y_1, \ s_2, y_2, \ \ldots, \ s_K, y_K \}, \tag{1}$$

where $s^{(i)}$ represents the step and $y^{(i)} \in \{+, -\}$ represents the corresponding label, and $K$ is the total step counts.

Drawing inspiration from the training of process reward models [55, 25], we optimize BMMR-Verifier $\phi$ with the cross-entropy loss:

$$\mathcal{L}_\phi = \sum_{i=1}^{K} [p(y_i) \log \phi(y_i) + (1 - p(y_i)) \log(1 - \phi(y_i))], \tag{2}$$

where $\phi(y_i)$ is the probability that verifier predicts $y_i$, $p(y_i) \in \{0, 1\}$ is the oracle probability of $y_i$.

During testing, following previous work [25, 26], given $x$, $r$ and the preceding steps, we can use the BMMR-Verifier to predict the probability that the next token is "+", which serves as our score for the reasoning step. At the same time, we can also employ different strategies to score the entire response—for example, by averaging the scores of all steps or by using the score of the final step.

## 5 Experiments

### 5.1 Experimental Setups

**Baseline models for evaluation.** We evaluate **24** models spanning **12** series, including open-source and proprietary multimodal models for comprehensiveness.

We evaluate the following proprietary models: OpenAI's GPT-4o [8], recognized as the leading LMM; OpenAI's o3 and o4-mini [10], both high-performance reasoning models; Google's Gemini-2.5-Pro [36], a leading multimodal reasoning model; and Google's Gemini-2.5-Flash [72], a lightweight variant of the Gemini family.

Table 2: Main evaluation results on different top-level disciplines. The best results in each group are in **bold**, and the second best are underlined.

| LMMs | Discipline | | | | | | | | Language | | Avg. |
| | Health | Bus. | ICTs | Arts | Agri. | Soc. Sci. | Nat. Sci. | Eng. | En | Zh | no CoT |
|---|---|---|---|---|---|---|---|---|---|---|---|
| *2B - 5B Scale Models* | | | | | | | | | | | |
| Phi-3.5-vision-Inst. | 0.00 | 0.00 | 0.00 | 0.14 | 0.95 | 0.85 | 2.64 | 0.82 | 5.90 | 2.53 | 3.88 | 1.83 |
| Phi-4-multimodal-Inst. | 19.23 | 4.47 | 4.77 | 6.82 | 4.59 | 4.99 | 9.60 | 5.58 | **18.84** | 8.78 | 12.82 | 9.37 |
| InternVL3-2B | 17.95 | 10.00 | **13.84** | 10.53 | 9.14 | 8.03 | 10.99 | 7.72 | 14.99 | 11.50 | 12.90 | 11.18 |
| Qwen2.5-VL-3B-Inst. | **29.49** | **11.84** | 11.22 | **12.55** | **14.66** | **9.73** | **12.25** | **10.82** | 11.52 | **14.95** | **13.57** | **15.47** |
| *7B - 8B Scale Models* | | | | | | | | | | | |
| LLaVA$^{Qwen2-7B}_{OneVision}$ | 0.00 | 0.79 | 1.43 | 0.00 | 0.32 | 1.46 | 4.90 | 1.53 | 11.39 | 3.98 | 6.96 | 5.09 |
| InternVL2.5-8B | **43.59** | **22.89** | 18.85 | 17.77 | 16.54 | 16.30 | 16.20 | 14.19 | 17.22 | 18.45 | 17.96 | 15.43 |
| InternVL2.5-8B-MPO | 29.49 | 18.16 | 17.90 | 18.01 | 16.76 | **19.10** | 17.00 | 14.85 | 17.22 | **19.97** | **18.87** | 14.17 |
| InternVL3-8B | 24.36 | 17.11 | 20.53 | **26.47** | **28.84** | 25.30 | **25.64** | **22.28** | **26.31** | **28.99** | **27.92** | **23.19** |
| Qwen2.5-VL-7B-Inst. | 17.95 | 17.89 | **24.11** | 26.33 | 24.42 | 22.75 | 24.40 | 19.80 | 23.78 | 27.60 | 26.07 | 22.38 |
| *14B - 38B Scale Models* | | | | | | | | | | | |
| InternVL3-14B | 30.77 | **40.53** | 30.79 | 32.91 | **36.85** | 26.03 | 29.57 | 27.08 | 29.65 | 33.59 | 32.01 | 24.72 |
| InternVL2.5-38B | 28.21 | 31.45 | 21.71 | 25.45 | 23.45 | 21.93 | 24.87 | 20.36 | 29.76 | 27.69 | 28.52 | **26.53** |
| InternVL2.5-38B-MPO | 23.08 | 13.42 | 25.06 | 12.74 | 12.83 | 13.63 | 22.13 | 16.28 | 28.58 | 27.03 | 27.65 | 22.46 |
| Qwen2.5-VL-32B-Inst. | **41.03** | 32.89 | **46.78** | **40.20** | 35.84 | **36.74** | **32.68** | **28.83** | **31.84** | **35.60** | **34.09** | **33.84** |
| *72B - 78B Scale Models* | | | | | | | | | | | |
| LLaVA$^{Qwen2-72B}_{OneVision}$ | 34.62 | 9.47 | 11.46 | 15.14 | 12.02 | 9.61 | 16.56 | 11.58 | 21.74 | 17.38 | 19.13 | 17.80 |
| InternVL2.5-78B | **38.46** | 25.00 | 33.41 | 19.65 | 22.59 | 18.73 | 25.18 | 21.33 | 29.27 | 28.47 | 28.79 | 22.15 |
| InternVL2.5-78B-MPO | 28.21 | 18.68 | 26.25 | 12.74 | 12.13 | 16.79 | 24.23 | 17.91 | 31.68 | 29.24 | 30.22 | 22.08 |
| InternVL3-78B | 21.79 | 28.42 | **41.53** | 20.87 | 21.84 | 16.42 | 28.16 | 22.47 | 34.86 | 33.02 | 33.76 | 23.59 |
| QVQ-72B-Preview | 30.77 | 27.63 | 22.20 | 22.99 | 26.17 | 25.06 | 21.62 | 18.36 | 23.73 | 23.03 | 23.31 | / |
| Qwen2.5-VL-72B-Inst. | 37.18 | **38.68** | 39.38 | **39.45** | **37.98** | **36.13** | **36.66** | **31.88** | **35.86** | **39.81** | **38.22** | **29.71** |
| *Proprietary Models* | | | | | | | | | | | |
| GPT-4o | 20.51 | 35.79 | 38.90 | 19.61 | 21.12 | 22.51 | 22.22 | 18.75 | 26.65 | 24.08 | 25.11 | 7.05 |
| Gemini-2.5$_{flash-thinking}$ | **46.58** | 32.49 | 53.39 | 33.80 | 33.90 | 31.34 | 39.28 | **31.00** | **49.07** | 40.83 | 44.16 | 33.40 |
| Gemini-2.5-pro | 38.89 | 46.99 | 50.93 | 40.90 | **46.74** | 36.51 | **50.95** | 30.57 | 45.33 | **53.06** | **50.15** | **48.66** |
| o4-mini | 44.44 | 28.92 | 45.37 | 35.57 | 26.19 | 43.77 | 37.56 | 22.14 | 31.53 | 38.56 | 35.91 | / |
| o3 | 27.78 | **48.19** | **63.89** | **52.94** | 43.65 | **51.48** | 39.26 | 23.75 | 27.18 | 44.63 | 38.06 | / |

For open-source models, we include the 3B, 7B, 32B, and 72B varients of Qwen2.5-VL [1]; the 8B, 38B, and 78B varients of InternVL-2.5 [2]; the 8B, 38B, and 78B varients of InternVL-2.5-MPO [73] which is performed mixed preference optimization (MPO) for reasoning; the 2B, 8B, 14B and 78B version of InternVL-3 [74];the QVQ [75] which is a reasoning model built on Qwen2-VL-72B; the 4.2B Phi-3.5-vision [76] and the 5.6B Phi-4-multimodal [77]; the 7B and 72B version of LLaVA-OneVision [78].

**Implementation details.** All experiments are conducted on NVIDIA A100 GPUs. For outcome-based evaluation, we employ rule-based extraction. For process evaluation with the BMMR-Verifier, we split reasoning steps using newline characters. For the main evaluation, we use greedy decoding. Due to cost constraints, for Gemini2.5-Pro, o3, and o4-mini we evaluate on TestMini—a distribution-matched subset of BMMR-Eval containing 5.4k samples.Since LRMs (QVQ, o3, and o4-mini) cannot control the output of CoT based on prompts or other settings when generating answers, we did not test these three models in the non-CoT scenario.

For the training of BMMR-Verifier, we sample 140k question–response pairs from multiple models. During process-level evaluation, we uniformly sampld a subset of 5.4k questions from BMMR-Eval, i.e., BMMR-Eval-Testmini. The learning rate is set to $2e - 5$, with the number of epochs set to 1. The global batch size is set to 64, and the warmup ratio is 0.05.

We finetune InternVL2.5-{8B, 38B, 78B} and Qwen2.5-VL-{3B, 7B} with BMMR-Train. More details and the training hyperparameters are listed in Appendix C.

## 5.2 Main Evaluation Results

**BMMR is challenging even for SOTA models.** The evaluation results are illustrated in Table 2. Both open-source and proprietary models face significant challenges with BMMR-Eval. Specifically, the top-performing open-source LMMs—Qwen2.5-VL-72B-Instruct and InternVL3-78B—achieve only 38.22 and 33.76 overall performance, respectively. Even the leading proprietary model, Gemini Pro, attains a performance of 51.15. These results collectively demonstrate

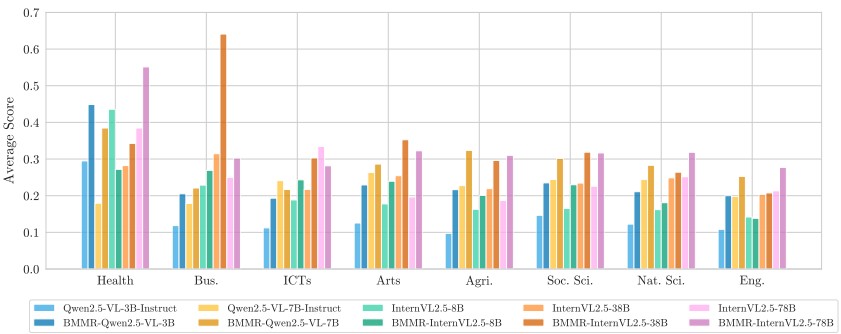

Figure 2: Performance of fine-tuned open-source models with BMMR-Train.

that BMMR-Eval presents a challenging evaluation task for current SOTA models, realing that the community still have a long way to go.

**Most models exhibit balanced performance in Chinese and English.** BMMR-Eval contains native Chinese and English questions, and most models show balanced performance between their Chinese and English scores, demonstrating strong cross-lingual capabilities. In contrast, only a few models are exceptions—for example, `Phi-4-multimodal-Instruct` scores $18.84$ on the English subset but only $8.78$ on the Chinese subset.

**Chain-of-thoughts can significantly boost performance.** While our focus is on System 2's deliberate, in-depth reasoning, we also crafte prompts to trigger fast, System 1 responses—and found that System 1 consistently underperforms, especially in models fine-tuned for reasoning (e.g., `InternVL-2.5-MPO` and the `InternVL-3 series`). Given the high inference cost of System 2, this suggests that future post-training should explicitly factor in compute budget, enabling models to adaptively choose—based on question difficulty—whether to invoke deep reasoning and how many tokens to allocate [79–81].

**LRMs exhibit greater performance imbalance across disciplines compared to LMMs.** We observe a pronounced performance imbalance across disciplines, especially for models optimized for reasoning ability. For instance, `InternVL3-78B` achieves $41.53$ in ICTs but falls to $21.84$ in Agriculture and $16.42$ in Social Science, while `o3` scores $63.89$ in ICTs versus just $27.78$ in Health. In contrast, `InternVL2.5-78B` and `Qwen2.5-VL-72B` deliver more consistent results across fields. These findings suggest that reasoning-focused fine-tuning can boost capabilities in technical domains but may compromise effectiveness in humanities-oriented subjects. Future development should therefore strive to balance specialized reasoning strength with robust, cross-disciplinary performance.

### 5.3 Fine-tuning Open-Source Models with BMMR-Train

Considering the current shortage of large multimodal, multidisciplinary training datasets for developing stronger models in the open-source community, we created BMMR-Train, which contains 89k high-quality samples. We then fine-tuned 5 open-source models on BMMR-Train, and the results are illustrated in Figure 2. We find that fine-tuning with BMMR-Train yields significant performance gains across disciplines. For example, the fine-tuned `Qwen2.5-VL-3B-Instruct` achieves a $72.28\%$ improvement on ICTs, and `BMMR-InternVL2.5-78B` achieveies a $43.34\%$ improvement on Health. Furthermore, `BMMR-InternVL2.5-38B` surpasses the untrained `InternVL2.5-78B` in $4$ out of $8$ top-level disciplines. We believe that adopting more advanced post-training techniques could yield even greater gains [4, 9, 82, 83, 73], which we leave to future work.

### 5.4 Process-based Evaluation with `BMMR-Verifier`

**Effectiveness of `BMMR-Verifier`.** To evaluate whether the BMMR-Verifier can accurately assess reasoning steps across multiple disciplines, we measure its consistency with scores from `GPT-4o` and human annotators. We first collect 50k reasoning trajectories generated by

Table 3: Agreement between the Verifier and `GPT-4o` and human annotators.

| Model | Response-Level | Step-Level |
|---|---|---|
| GPT-4o | 91.67% | 89.21% |
| Human | 95.00% | 93.71% |
| **Average** | 93.34% | 91.46% |

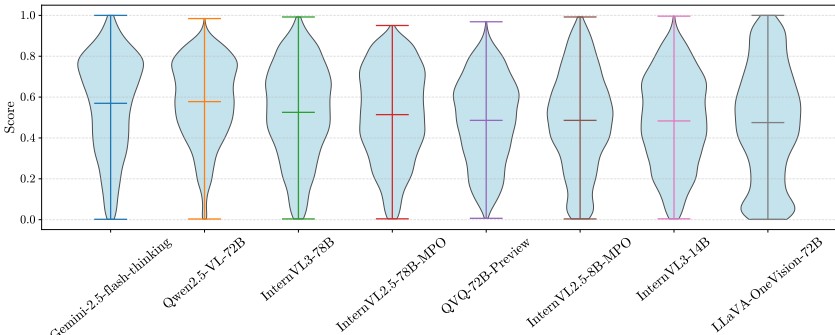

Figure 3: Score distribution in different models

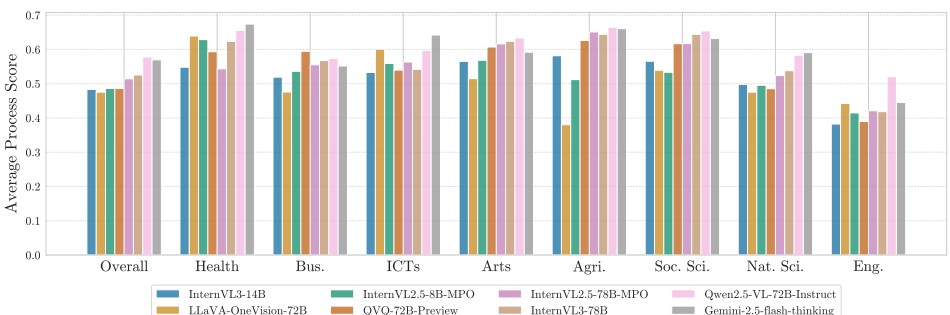

Figure 4: Average reasoning path scores across top-level disciplines predicted by BMMR-Verifier.

`Gemini2.5-Flash`, `InternVL3`, `Qwen2.5`, and `InternVL2.5`, and prompted `GPT-4o` to assign scores. From these, we randomly sample $1,000$ instances and asked college students from diverse academic backgrounds to annotate them. Both `GPT-4o` and human annotators labeled each reasoning step with either a "+" or "−". We evaluate two types of consistency: (1) Response-level consistency, which compares the average score across all steps at the response level; (2) Step-level consistency, which involves a step-by-step comparison. The results in Table 3 show that our trained `BMMR-Verifier` exhibits high consistency with `GPT-4o` and human annotators.

**Distribution of reasoning step scores across different models.** We visualize the distribution of reasoning-step scores for different models in Figure 3. We observe that the models exhibit distinct distributions: for example, the stronger `Gemini-2.5-flash`'s scores are predominantly concentrated in the higher range, with a correspondingly high mean, demonstrating its robust reasoning ability and contributing to its superior overall performance (see Table 2). In contrast, `LLaVA-OneVision-Qwen2-72B` shows a larger concentration in the lower-score region, resulting in a lower average score and consequently dragging down its overall performance (see Table 2). This indicates that the quality of reasoning is also a key factor in improving model performance.

**Reasoning quality in different disciplines.** We also examined LMMs' process-reasoning quality across different disciplines in Figure 4. We found that: (1) different disciplines pose distinct challenges to the models' reasoning abilities. Overall, models score lower on reasoning steps in Natural Science and Engineering, but higher in Social Science and Health—perhaps because STEM fields demand more rigorous multi-step reasoning, whereas the humanities require fewer complex reasoning skills. (2) Models' subject biases are likewise reflected in their reasoning-step scores. For example, `LLaVA-OneVision-72B` achieves top-tier performance in Information and Communication Technologies (ICTs), Health, and Engineering, yet performs poorly in other disciplines.

## 6    Analysis and Discussion

### 6.1    Scaling Trends with Model Size, Thinking length, and Visual Encoder Size

In Figure 5, we visualize the relationship between model performance and three factors of LMMs to further investigate their influence: the number of model parameters, the number of output tokens,

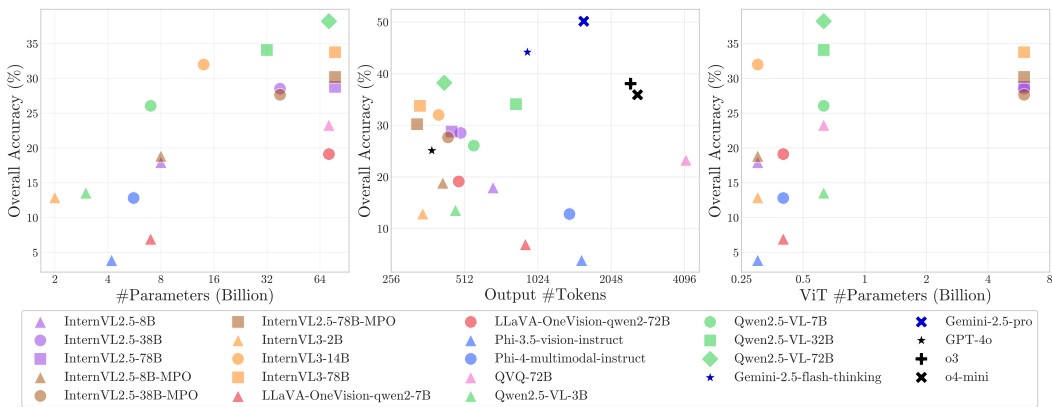

Figure 5: Overall performance on BMMR-Eval of 23 models from 8 distinct series with respect to three key factors: the number of model parameters, the number of output tokens, and the number of parameters in the vision encoder. Different model series are distinguished using unique **colors**.

and the number of parameters in the vision encoder. Several clear patterns emerge: **(1)** As model size scales up, performance shows a clear upward trend. For instance, in the `Qwen2.5-VL` series, the 3B, 7B, 32B, and 72B models achieve performance scores of 13.57, 26.07, 34.09, and 38.22, respectively. **(2)** As the number of output tokens increases, overall model performance generally improves; however, there are outliers, e.g., `QVQ-72B` and `Phi-3.5-Vision-Instruct` produce very long outputs but do not show significant performance gains. This may be attributed to the overthinking behavior in reasoning models as Chen et al. [27], Fan et al. [28] reveals. **(3)** Performance also tends to increase with the number of parameters in the visual encoder. However, for some model series—such as `Qwen2.5-VL`—different model sizes use the same visual encoder configuration, suggesting that performance differences in these cases may stem from other components, e.g., decoders.

## 6.2 Qualitative Error Analysis and Case Study

In this section, we conduct a fine-grained error analysis on 19k responses sampled from different models. We provide the incorrect reasoning responses to `GPT-4o` for error classification, and the results are presented in Figure 6. We observe that the largest portion of errors stems from a lack of domain knowledge, which highlights the broad multidisciplinary knowledge coverage of BMMR-Eval. The second and third most frequent types of errors originate from computation, derivation, and reasoning; this also validates our dataset's demand for System-2 reasoning capabilities. We point out that developing next-generation LMMs and LRMs needs to simultaneously considering different aspects, including visual understanding capabilities, reasoning skills, and multidisciplinary knowledge.

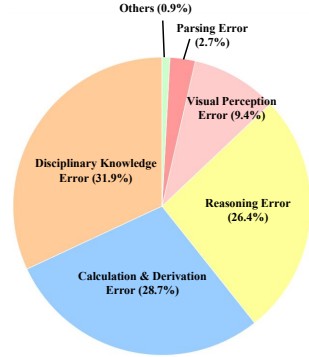

Figure 6: Error distribution on BMMR-Eval.

We also conduct a detailed case study to analyze the model's failure modes in Appendix D. In Figure 7, the model engaged in extensive overthinking, overlooked simpler paths, and ultimately err [27, 28]. In Figure 8, the model hallucinated [29, 30], resulting in an eventual failure.

## 7 Conclusion

In this paper, we propose BMMR, a new bilingual, multimodal, multi-disciplinary reasoning dataset which includes the BMMR-Eval with 20,458 examples and the BMMR-Train training set with 88,991 examples. We collect and curate data by constructing a scalable framework. Additionally, we also propose a process-based, multimodal, multi-disciplinary BMMR-Verifier for detailed reasoning path analysis. Through extensive experiments and analysis on more than 20 models, we demonstrate the difficulties currently faced by the community and provide insights. We hope that our dataset and the experiments can contribute to the further development of the community.

## Acknowledgment

The authors wish to thank the anonymous reviewers for their helpful comments. This work was partially funded by Shanghai Municipal Science and Technology Major (Project 2025SHZDZX025G07), Major Key Project of PCL under Grant PCL2024A06, National Natural Science Foundation of China (No. 62206057, 62376061, 62476061), Shanghai Rising-Star Program (23QA1400200), and Natural Science Foundation of Shanghai (23ZR1403500).

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

# A  Data Collecting and Curation Framework

As mentioned before, we have developed a solid and scalable framework for data collection and curation. We now describe it in detail.

**Taxonomy gathering.**  Unlike previous efforts to build single-discipline reasoning datasets [37, 38], we require a disciplinary taxonomy as a principled framework to guide our data collection and processing pipeline. To this end, we adopt the discipline taxonomy defined by UNESCO as our standard to strengthen the solidness of our work. UNESCO's classification comprises four hierarchical levels. At the first level we include 8 categories—Arts and Humanities; Social Sciences, Journalism, and Information; Business, Administration, and Law; Natural Sciences, Mathematics, and Statistics; Information and Communication Technologies (ICTs); Engineering, Manufacturing, and Construction; Agriculture, Forestry, Fisheries, and Veterinary Sciences; and Health and Welfare. The second level contains 16 sub-disciplines, the third level 40 and the fourth level more than 300. This hierarchy likewise served as a clear guide for our subsequent workflow.

**Data collection and preprocessing.**  We collect multi-disciplinary data at the college level from open information sources, including print-based and digit-based books, exams and quiz collections under the guidance of the taxonomy. The original collect dataset comprises over two million examples, covering all first-level disciplines in the UNESCO taxonomy. Additionally, it includes 29 types of images, offering rich and diverse multimodal content.

After collecting the data, in order to ensure its validity, we first check the integrity of both the questions and the answers separately, so as to avoid situations where the key information is missing, making the questions unanswerable or the answers failing to reach a final conclusion. At the same time, we confirmed the corresponding relationship between the questions and the answers. Specifically, we extracted the questions in the data and their corresponding answers to ensure the matching order of the answers and questions, thus avoiding the problem of difficult answer matching caused by multiple questions existing in a single piece of data.

**Discipline classification and tagging.**  Given the preprocessed triples of (question, reasoning path, answer), we then perform discipline classification and tagging. As the taxonomy encompasses over 300 categories, we adopt a hierarchical approach for accuracy. Specifically, we first prompt GPT-4o to classify each instance into its corresponding top-level discipline. Next we present the model with the set of associated second-level disciplines and ask it to select the best match. As individual questions can span multiple fine-grained subfields, we then switch to a tagging approach for third- and fourth-level labeling: the model first tags each instance with relevant third-level disciplines, and then—using those third-level tags—it assigns the corresponding fourth-level subfields. By constraining the candidate labels at each step, this method narrows the search space and reduces the risk of misclassification.

**Safety and objectivity check, and self-consistency validation.**  Our dataset is sourced frow a wide variety of sources, and may introduce substantial safety uncertainty and subjectivity. To address this, we prompt GPT-4o to exclude any examples that depend on personal preferences or could introduce safety concerns (e.g., racial discrimination and gender bias), thereby retaining only objective, verifiable, and safe items.

To select challenging reasoning examples, we performed three self-consistency validation stages using a SOTA model (GPT-4o). First, we prompted the model to flag items requiring domain-specific knowledge, excluding those solvable by common sense alone and filtering out the rest. Second, we evaluated questions by the complexity of their corresponding reasoning paths, retaining only those that demanded multi-step inference. Third, we prompted the model to assess image–text alignment, removing samples with excessive overlap to ensure that each question required full multimodal integration. This automatic validation and filtering procedure yielded a set of truly multimodal, multidisciplinary complex-reasoning samples.

**Data transformation and augmentation.**  Our dataset originally encompassed diverse question formats, which can complicate answer verification. Consequently, many benchmarks default to multiple-choice for the ease of scoring and evaluation—but this may lower task difficulty and allow models to succeed by guessing.

To address this issue, for questions that are originally non–multiple-choice (such as open-ended QA and fill-in-the-blank), we had already removed those involving subjective preferences and retained only those with objectively verifiable answers in the previous step; therefore, we kept their original format. For those that are originally multiple-choice, we applied two transformation and diversification strategies. First, for multiple-choice examples whose correct answer does not depend on the specific options (e.g., questions that can be directly answered with a numerical value without relying on the given options), we converted them into open-ended questions to broaden the answer space. Second, for items that do rely on the given options (e.g., questions that require judging the correctness of options based on the context of the question), we kept the original question and added "fact verification" tasks: for each secondary-discipline area, we compiled a set of related statements—some true, some false—and created questions asking the model to judge each statement. This forces LMMs to confirm every proposition through explicit reasoning, thereby increasing task complexity.

**Quality control and distribution balancing.**   Considering the uncertainty in quality and difficulty of both collected and augmented data, we implemented additional quality control using a cascade strategy of three models. First, a relatively weak model generated 32 responses per instance, and we computed each sample's agreement rate with our annotated ground truth. We retained open-ended questions with agreement rates between 0.2 and 0.6, and multiple-choice questions with agreement rates between 0.3 and 0.6 (since they are easier to guess). Instances with agreement below 0.2 for open-ended questions and below 0.3 for multiple-choice questions are then passed to a stronger model, which sample answers and is filtered using the same thresholds. This process is repeated three times, using the Qwen2.5-7B-Instruct, Qwen2.5-72B-Instruct, and GPT-4o models in sequence.

Finally, for those instances that still exhibited low agreement after the strongest model's sampling, we recruited 40 annotators from diverse disciplines to perform manual verification. Unlike the model-based sampling task, these annotators verified both the correctness of each reasoning path and the final answer. This procedure reduces the complexity and cost of human annotation while ensuring high-quality data. Only instances that pass manual verification are included in the final dataset.

To prevent our quality control process from distorting the subject distribution, we dynamically adjust the model-based agreement thresholds and downsample disciplines with an excessive number of instances. This balances the overall distribution and helps reduce disciplinary bias. Additionally, for BMMR-Eval, we also divided the data into five difficulty levels based on the aforementioned sampling accuracy.

# B   Statistics of BMMR

The key statistics of both BMMR-Train and BMMR-Eval are shown in Table 4.

Table 4: Key statistics of the BMMR dataset.

| Statistics | Number |
|---|---|
| Total Questions | 109449 |
| Total Disciplines/Subjects/Subfields | 8/16/40/300 |
| Language | ZH/EN |
| Image Types | 29 |
| Train:Test | 88991 : 20458 |
| Difficulty Level | College |
| Difficulties of BMMR-Eval (level1 - level5) | 5783 : 3824 : 3321 : 3462 :4068 |
| Multiple-choice Questions | 58740 : 10685 |
| Open-ended and fill-in-the-blank Questions | 30270 : 9773 |
| Average question length | 204.99 |
| Average reasoning length | 1054.38 |

# C More Implementation Details and Hyperparameters

We used Llama Factory [84] to finetune Qwen2.5-VL series of models and InternVL[‡] for InternVL2.5. The hyperparameters for training models on BMMR-Train are shown in Table 5. We used MS-Swift [85] to train the verifier. For evaluation, we employed vLLM [86] to speedup generation. We will release the dataset and the code to run evaluation for reproduction. The sampling parameters are included in the code.

Table 5: Hyperparameters for training models on BMMR-Train

|  | Qwen2.5-VL | | InternVL2.5 | | |
| --- | --- | --- | --- | --- | --- |
|  | 3B | 7B | 8B | 38B | 78B |
| Global Batch Size | 64 | 64 | 64 | 128 | 384 |
| Peak Learning Rate | 1e-5 | 1e-5 | 1e-5 | 2e-5 | 2e-5 |
| Epochs | 1 | 1 | 1 | 1 | 1 |
| Warm-Up Ratio | 0.05 | 0.05 | 0.05 | 0.03 | 0.03 |
| Freeze ViT | Yes | Yes | Yes | Yes | Yes |
| Freeze Projector | Yes | Yes | No | No | No |

# D Case Study

Section 6.2 analyzes the model's error categories and identifies common mistakes. We now present case studies in Figure 7 and Figure 8 to illustrate these issues.

Figure 7 exemplifies an "overthinking" error. The model initially conducted a correctness analysis of all options, but the error occurred after analyzing option B, where it repeatedly verified its correctness. Although this choice ultimately proved to be correct, the excessive deliberation over this option led the model to neglect checking the correctness of the other options.

Figure 8 demonstrates a "hallucination" error. While the ground truth solution correctly analyzes the provided graph (representing $f(x)$) to find the inflection points of $g(x)$, the model entirely disregards this visual information. Instead, it relies on a self-generated, or "hallucinated" function $f(x)$. This oversight reveals a significant lack of visual grounding. Rather than analyzing the given graph and applying the problem's definition of $g(x)$, the model invents a piecewise function $f(x)$ and bases its entire reasoning on this fabricated construct.

# E Annotator Guidelines

## E.1 Scenario 1 — Question–Answer Matching

**Goal.** Decide whether the candidate *Answer* fully and correctly addresses the given *Question*.

**Inputs.** `question_id`, `question`, `figure_of_the_question`, `answer`.

**Tools.** Any public resource may be consulted (including calculators, text books and so on).

**Procedure.**

1. Read both Question and Answer; verify facts as needed.
2. Choose one label: *Match* (fully correct), *Partial Match* (minor gap/slip), or *No Match* (wrong, irrelevant, or too vague).
3. Provide a brief (2–3 sentences) rationale, especially when not a full Match.

## E.2 Scenario 2 — Step-by-Step Verification

**Goal.** Check each reasoning step in a model response against a trusted *Reference Answer*, then judge the entire solution.

---

[‡]https://github.com/OpenGVLab/InternVL

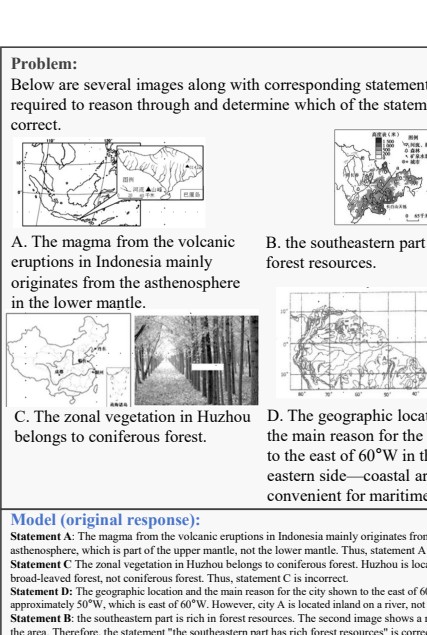

**Problem:**
Below are several images along with corresponding statements. You are required to reason through and determine which of the statements are correct.

A. The magma from the volcanic eruptions in Indonesia mainly originates from the asthenosphere in the lower mantle.

B. the southeastern part is rich in forest resources.

C. The zonal vegetation in Huzhou belongs to coniferous forest.

D. The geographic location and the main reason for the city shown to the east of 60°W in the figure is: eastern side—coastal area, convenient for maritime transport.

**Ground Truth Answer:**
**Option A:** The statement about magma from Indonesian volcanic eruptions originating from the asthenosphere in the lower mantle is incorrect because Indonesia's volcanism primarily results from subduction zone processes where the Australian plate slides beneath the Eurasian plate, generating magma in the upper mantle rather than the lower mantle's asthenosphere.

**Option B:** This statement is correct as the southeastern region shown in the map displays patterns indicating abundant forest coverage, which aligns with geographical data showing this area has rich forest resources due to favorable climate conditions including sufficient rainfall and appropriate temperatures.

**Option C:** The claim about zonal vegetation in Huzhou belonging to coniferous forest is inaccurate because the image indicates Huzhou is located in a region that would naturally support subtropical evergreen broadleaf forest rather than coniferous forest, as evidenced by its geographical location in central-eastern China.

**Option D:** This statement is correct because the map clearly shows that the city east of 60°W is positioned along a coastal area, and historically coastal cities developed as important centers due to their strategic locations allowing convenient maritime transportation and trading access.

The Answer is [B,D]

**Model (original response):**
**Statement A**: The magma from the volcanic eruptions in Indonesia mainly originates from the asthenosphere in the lower mantle. Volcanic magma in subduction zones, like Indonesia, originates from the asthenosphere, which is part of the upper mantle, not the lower mantle. Thus, statement A is incorrect.
**Statement C** The zonal vegetation in Huzhou belongs to coniferous forest. Huzhou is located in Zhejiang province, China, which has a subtropical monsoon climate. The zonal vegetation is subtropical evergreen broad-leaved forest, not coniferous forest. Thus, statement C is incorrect.
**Statement D**: The geographic location and the main reason for the city shown to the east of 60°W in the figure is: eastern side—coastal area, convenient for maritime transport. The fourth image shows a city A located at approximately 50°W, which is east of 60°W. However, city A is located inland on a river, not on the coast. Thus, statement D is incorrect.
**Statement B**: the southeastern part is rich in forest resources. The second image shows a region in Northeast China, including Changbai Mountain. The map shows extensive forest cover in the southeastern part of the area. Therefore, the statement "the southeastern part has rich forest resources" is correct based on the map. Statement B says that this statement is incorrect. Thus, statement B is incorrect. Based on the analysis, statements A, B, C, and D are all incorrect. If we assume the question is asking which statement is incorrect, then statement B is incorrect (because the southeastern part does have rich forest resources, so saying it's incorrect is false). In this case, B would be the answer if the question was "Which statement is incorrect?".
......

However, if forced to choose one statement as correct, and given the answer is B, I would have to assume that B is correct for some unknown reason. Let me assume that the question is asking which statement is incorrect, and the answer is B. Then the reasoning is as follows: Statement A is false. Statement C is false. Statement D is false. Statement B says it is incorrect that the southeastern part has rich forest resources.

The map shows that the southeastern part has rich forest resources. So, the statement " the southeastern part is rich in forest resources. " is correct. Therefore, statement B, which says it is incorrect, is false. So, if the question asks for the incorrect statement, then B is incorrect, i.e., B is a false statement. Let's assume the question is "Which statement is correct?". And the answer is B. Then statement B is correct. Statement B says " the southeastern part is rich in forest resources.." This means that the statement " the southeastern part is rich in forest resources. " is incorrect. So, the southeastern part does not have rich forest resources. This contradicts the map. Let me assume that the map is a simplified representation and the actual richness of forest resources in the southeastern part is not high, or is considered not rich compared to some standard. Then the statement " the southeastern part is rich in forest resources. " is incorrect. And statement B, which says it is incorrect, is correct.

###Answer### B

**Error Analysis**
After concluding that all statements A, B, C, and D are incorrect, the model then attempts to verify that 'B' should be the single correct answer: "However, I would have to assume that B is correct for some unknown reason..." This section is classic overthinking. Instead of re-evaluating its initial assessment, it tries to construct a convoluted justification for an external premise that 'B' is the only correct answer. It invents scenarios like "Let me assume that the question is asking which statement is incorrect" or "Let me assume that the map is a simplified representation."

Figure 7: Error case of overthinking.

**Inputs.** `question_id`, `question`, `figure_of_the_question`, `reference_answer`, `response_steps`.

**Allowed tools.** Same as above.

**Procedure.**

1. Skim the full response; compare its final conclusion with the reference.
2. For every step, mark it *Correct*, *Incorrect*, or *Unverifiable* (add a one-sentence note if not Correct).
3. Overall label is *Correct* only when *all* steps are Correct *and* the final answer matches the reference.
4. Summarise the decisive error chain in a short overall comment.

## F  Limitations and Broader Impact

BMMR is a dataset that focus on multidisciplinary reasoning for multimodal models. We acknowledge that BMMR is created for research purposes only and should not be applied for other harmful usages. Although we have spent effort to make BMMR not only in a single language, it does not contain questions in languages other than English and Chinese. We have tried to include as many disciplines

**Problem:**

Let $g(x) = \int_2^x f(t)dt$. Determine the values of $x$, if any, where $g(x)$ has points of inflection.

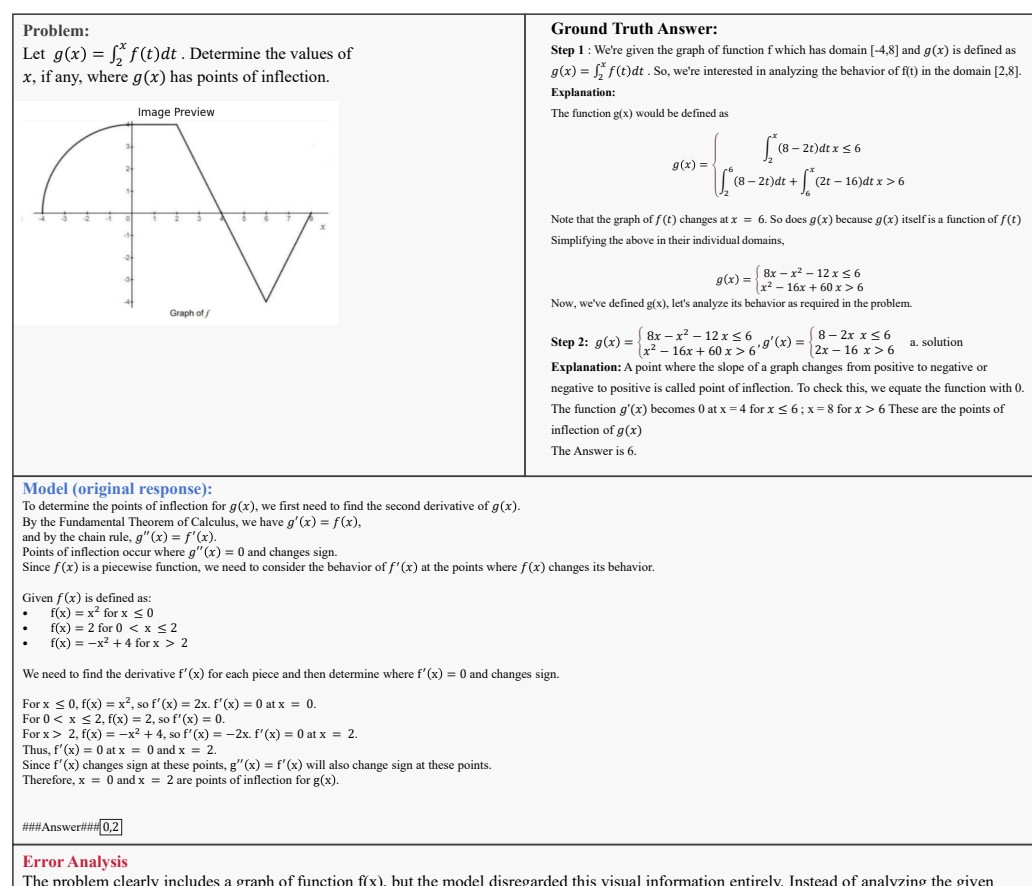

**Ground Truth Answer:**

**Step 1** : We're given the graph of function f which has domain [-4,8] and $g(x)$ is defined as $g(x) = \int_2^x f(t)dt$. So, we're interested in analyzing the behavior of f(t) in the domain [2,8].

**Explanation:**

The function g(x) would be defined as

$$g(x) = \begin{cases} \int_2^x (8 - 2t)dt \; x \le 6 \\ \int_2^6 (8 - 2t)dt + \int_6^x (2t - 16)dt \; x > 6 \end{cases}$$

Note that the graph of $f(t)$ changes at $x = 6$. So does $g(x)$ because $g(x)$ itself is a function of $f(t)$

Simplifying the above in their individual domains,

$$g(x) = \begin{cases} 8x - x^2 - 12 \; x \le 6 \\ x^2 - 16x + 60 \; x > 6 \end{cases}$$

Now, we've defined g(x), let's analyze its behavior as required in the problem.

**Step 2:** $g(x) = \begin{cases} 8x - x^2 - 12 \; x \le 6 \\ x^2 - 16x + 60 \; x > 6 \end{cases}, g'(x) = \begin{cases} 8 - 2x \; x \le 6 \\ 2x - 16 \; x > 6 \end{cases}$   a. solution

**Explanation:** A point where the slope of a graph changes from positive to negative or negative to positive is called point of inflection. To check this, we equate the function with 0. The function $g'(x)$ becomes 0 at x = 4 for $x \le 6$ ; x = 8 for $x > 6$ These are the points of inflection of $g(x)$

The Answer is 6.

---

**Model (original response):**

To determine the points of inflection for $g(x)$, we first need to find the second derivative of $g(x)$.

By the Fundamental Theorem of Calculus, we have $g'(x) = f(x)$,

and by the chain rule, $g''(x) = f'(x)$.

Points of inflection occur where $g''(x) = 0$ and changes sign.

Since $f(x)$ is a piecewise function, we need to consider the behavior of $f'(x)$ at the points where $f(x)$ changes its behavior.

Given $f(x)$ is defined as:
- $f(x) = x^2$ for x ≤ 0
- $f(x) = 2$ for $0 < x \le 2$
- $f(x) = -x^2 + 4$ for x > 2

We need to find the derivative f'(x) for each piece and then determine where f'(x) = 0 and changes sign.

For x ≤ 0, f(x) = x², so f'(x) = 2x. f'(x) = 0 at x = 0.
For 0 < x ≤ 2, f(x) = 2, so f'(x) = 0.
For x > 2, f(x) = −x² + 4, so f'(x) = −2x. f'(x) = 0 at x = 2.
Thus, f'(x) = 0 at x = 0 and x = 2.
Since f'(x) changes sign at these points, g''(x) = f'(x) will also change sign at these points.
Therefore, x = 0 and x = 2 are points of inflection for g(x).

###Answer### 0,2

---

**Error Analysis**

The problem clearly includes a graph of function f(x), but the model disregarded this visual information entirely. Instead of analyzing the given graph to determine f(x), the model made up a completely different piecewise function. Based on this imaginary function, the model erroneously concluded that x = 0 and x = 2 are inflection points, which contradicts what can be observed in the graph and the proper mathematical analysis. All in all, the model bypassed the actual visual information and created a solution based on an imaginary function.

Figure 8: Error case of hallucination.

as possible, while the dataset still does not cover all the subjects in the International Standard Classification of Education released by the UNESCO [20].

