# OpenReview forum: "BMMR: A Large-Scale Bilingual Multimodal Multi-Discipline Reasoning Dataset"
_NeurIPS.cc/2025/Datasets_and_Benchmarks_Track — NeurIPS 2025 Datasets and Benchmarks Track poster_

### Official Review · Reviewer_NWzd · 2025-07-02

**Rating:** 4
**Confidence:** 4

**Summary:**

This paper extracts a dataset from existing UNESCO materials to create a bilingual, multimodal benchmark designed to test the reasoning abilities of models across various academic disciplines. The dataset is composed of a training set and a test set. The authors' experiments demonstrate that current models perform poorly on the test set, but their performance can be improved by training on the provided training set. To enable a more fine-grained, process-based evaluation of model outputs, the authors have also trained a companion verifier to facilitate detailed scoring. Based on these results, the authors provide an analysis of common error types.

**Dataset Code Accessibility:**

Yes

**Ethical Considerations:**

No, there are no or only very minor ethics concerns

**Final Justification:**

Rating: 4
This paper is worth a rating of 4, considering the responses provided by the authors.

**Limitations Weaknesses:**

1. The experimental setup for fine-tuning, as described in Section 5.3, is methodologically flawed. The authors perform continual training on models where the training and test sets are drawn from the same distribution. This approach risks having the LVLMs simply overfit to this specific data distribution, learning its inherent biases and noise rather than generalizable patterns.

2. A more robust methodology would be to mix the authors' training data with other visual instruction tuning datasets. The resulting models should then be evaluated not only on the proposed benchmark but also on a variety of other existing benchmarks. This would test for true generalization, rather than merely evaluating performance on an isolated, in-distribution test set.

**Strengths Contributions:**

1. The quality of the data appears to be good.

2. The research direction is commendable. The community is currently focused heavily on math reasoning while often neglecting other domains. This work helps address whether models possess general reasoning abilities or are merely overfitting to the mathematical domain.

3. The provision of a training set is a valuable contribution, as it can serve as a training resource for other LLMs and LVLMs.

---

> ### Author Rebuttal · Authors · 2025-07-31
>
> Thank you very much for your valuable time and feedback. Your suggestions regarding training and testing for generalization is extremely helpful, and we conducted relevant experiments to address your concerns!
>
> ---
>
> > Q1: Question about Overfitting:
>
> A1:
>
> - Following your insightful advice, we designed several training data combination strategies, including:
>
> 1. Using only BMMR-train (BMMR-80k)
>
> 2. Mixing 40k samples randomly selected from both BMMR-train and MMPR-v1.2 (BMMR-40k + MMPR-40k)
>
> 3. Mixing 40k samples randomly selected from BMMR-train with 80k samples randomly selected from MMPR-v1.2 (BMMR-40k + MMPR-80k)
>
> 4. Mixing the full BMMR-train (80k) with 80k samples from MMPR-v1.2 (BMMR-80k + MMPR-80k)
>
> - We conducted the above training experiments using InternVL2.5-4B. During training, the learning rate was set to 1e-5, the global batch size was 32, and the model was trained for 1 epoch. The ViT module was frozen throughout training.
>
> | Dataset           | MMMU | MathVerse (mini) - Text Lite | MathVerse (mini) - Vision Dominant | MathVerse (mini) - Vision Only | MathVerse (mini) - Vision Intensive | MathVerse (mini) - Text Dominant | AVG  |
> |-------------------|------|------------------------------|------------------------------------|--------------------------------|-------------------------------------|----------------------------------|------|
> | None              | 50.7 | 34.1                         | 31.3                               | 27.3                           | 31.0                                | 40.9                             | 41.81 |
> | BMMR-80k          | 53.3 | 35.3                         | 32.1                               | 28.7                           | 30.5                                | 42.8                             | 43.59 |
> | BMMR-40k+MMPR-40k | 51.3 | 36.0                         | 33.6                               | 31.3                           | 34.4                                | 44.3                             | 43.61 |
> | BMMR-40k+MMPR-80k | **54.7** | 37.3                         | 33.8                               | **31.5**                           | 31.7                                | 44.4                             | 45.22 |
> | BMMR-80k+MMPR-80k | 53.3 | **38.3**                         | **35.5**                               | 30.1                           | **35.3**                                | **47.6**                             | **45.33** |
>
> - To evaluate the model’s generalization in multi-disciplinary and mathematical domains, we tested its performance on two widely used benchmarks: MMMU and MathVerse. As shown in the results table, the model consistently improved across various tasks on both benchmarks. Even when using only BMMR-train, we observed improvements in almost all tasks, with an average increase of **1.78%**, demonstrating the strong generalization ability of our training set and its capacity to transfer multi-disciplinary knowledge to unseen domains.
>
> - Furthermore, as per your suggestion, by incorporating the open-source dataset MMPR and increasing in overall training data size, the model’s overall performance continued to improve. The average accuracy showed a steady upward trend, with the BMMR-80k + MMPR-80k setting achieving an **3.52%** increase in average accuracy.
>
> - It is worth noting that despite the absence of fine-grained parameter tuning during training, the model has already achieved promising results. We plan to further optimize training parameters and data mixing strategies in future work, and extend training to more models to more comprehensively validate the generalization of our training set.
>
> Thank you again for your feedback. Based on your suggestions, we have added additional experiments across different tasks to address your concerns. Please feel free to let us know if you have any further questions. If you find our revisions satisfactory, we kindly ask you to consider raising your score.

---

> > ### Author Response · Authors · 2025-08-05
> > **Friendly Reminder about the Discussion**
> >
> > Dear Reviewer NWzd,
> >
> > Thank you for your valuable time and great efforts in reviewing our work and for your insightful questions. As the deadline for the reviewer–author discussion period approaches, we have not yet received your response to our rebuttal. We hope that our answers have helped to clarify the points raised, and we look forward to hearing from you. Please let us know if you require any further information or have any additional concerns, and we will try our best to address them. Thank you again!
> >
> > Best regards,
> >
> > Authors of Submission 2221

---

> > ### Comment · Reviewer_NWzd · 2025-08-06
> >
> > I read the author's reply and decided to maintain the existing positive score.

---

> ### Author Response · Authors · 2025-08-06
> **Kind Invitation to Discussion**
>
> Dear Reviewer NWzd,
>
> Thank you again for your thoughtful and constructive reviews. We appreciate the time and effort you’ve dedicated to evaluating our work.
>
> May we kindly ask whether our rebuttal has adequately addressed your concerns? If there are any remaining questions or points you'd like us to clarify, we’d be happy to provide additional information or make further improvements.
>
> Best regards,
>
> Authors of Submission 2221

---

### Official Review · Reviewer_vsde · 2025-07-02

**Rating:** 5
**Confidence:** 4

**Summary:**

The paper introduces BMMR, a new large-scale, bilingual dataset designed for evaluating the reasoning capabilities of large multimodal models. The dataset contains 11k samples covering 30 subjects as defined by UNESCO. These questions are presented in various formats, including multiple-choice, fill-in-the-blank, and open question-answering, and are sourced from both print and digital media. Each question in the dataset is accompanied by a detailed reasoning path. Moreover, this paper introduces a verifier for accurate and fine-grained evaluation of the reasoning paths generated by models. The paper presents extensive experiments conducted on different models, including state-of-the-art systems like GPT-4o and Gemini-2.5-Pro. The results indicate that even the most advanced models have significant room for improvement when tested on the proposed dataset.

**Dataset Code Accessibility:**

Yes

**Ethical Considerations:**

No, there are no or only very minor ethics concerns

**Final Justification:**

I think the feedback has answered my questions well, and I give it a relatively positive rating.

**Limitations Weaknesses:**

While the dataset is bilingual, including more languages ​​could make the dataset more useful. The dataset is more focused on academic knowledge, but recent research has focused more on open-domain multimodal settings.

**Strengths Contributions:**

This paper introduces a large-scale and diverse dataset for evaluating the reasoning abilities of large multimodal models. This dataset is of great significance for the recent development of multimodal models. This paper provides an in-depth analysis of the performance of typical multimodal models, showing that there is a significant performance gap between existing models and our expectations.

---

> ### Author Rebuttal · Authors · 2025-07-31
>
> Thanks for your valuable review, feedback, and recognition of our work. This has given us great encouragement.
>
> ---
>
> > Q1： Suggestion about the introduction of more languages.
>
> A1:
>
> - Regarding our multi-disciplinary multimodal dataset, incorporating more languages would undoubtedly benefit a broader community. However, limited by our data acquisition channels, we constructed the dataset using only English and Chinese data.
> Despite this, we have conducted a detailed analysis of the performance of different models in Chinese and English, and provided insights such as **language bias** in different models.
>
> - For future language expansion, during the construction of our bilingual dataset, we developed **a comprehensive data processing pipeline** that can efficiently process new data into training or high-quality evaluation samples. Therefore, once we acquire data in new languages, **we can effectively extend our dataset to cover more languages and disciplines based on this pipeline**. This will provide an extensible pipeline for constructing and evaluating multimodal reasoning data across more languages.In addition, we can also translate many of questions to further expand our dataset. We will reserve these solutions for future work. Thank you again for your suggestions!
>
>
>
> > Q2: Suggestion to open-domain scenarios.
>
> A2:
>
> - Thank you for your suggestion. Currently, our work focuses on multidisciplinary reasoning, with the goal of **exploring the general capabilities of models across diverse academic fields**. In the future, considering the value of downstream applications (such as autonomous driving, embodied intelligence, etc.), we hope to extend our pipeline to more open domains, thereby **transforming it from an expert in multidisciplinary reasoning into a proficient in perception and decision-making in the real world**.
>
> Thank you again for your suggestion. If you have any other questions, please feel free to ask! We will carry out future work based on your suggestions, and we hope we can make more contributions to the community. If you are satisfied with our responses, we kindly ask you to adjust the score accordingly. Thanks again!

---

> > ### Author Response · Authors · 2025-08-05
> > **Thanks for Reviewer vsde's review! Authors' feedback**
> >
> > Thanks for your valuable time and great efforts in reviewing our work and for your insightful questions. We hope that our answers and responses have helped to clarify the points discussed. Please let us know if there is anything else you require further information on or if there are any additional concerns you might have.

---

### Official Review · Reviewer_txoN · 2025-07-03

**Rating:** 5
**Confidence:** 3

**Summary:**

This work introduces BMMR, a large-scale bilingual multimodal reasoning dataset comprising over 110K college-level questions across 300 UNESCO-defined subjects. To support fine-grained evaluation, the authors propose BMMR-Verifier, a multi-discipline framework for evaluate reasoning paths. Experimental results uncover that LLMs can further enhance their multimodal reasoning capabilities through post-training.

**Additional Feedback:**

1. Figure Readability: The fonts in several figures (e.g., Figure 1) are too small and should be bolded or enlarged to improve readability.
2. Line 246–247: The authors should specify the number of annotators involved in the labeling task to enhance transparency.
3. Line 67: The authors are encouraged to provide stronger justification or empirical evidence to support the claim that “overthinking” and “hallucination” are key challenges in multimodal reasoning.

**Dataset Code Accessibility:**

Partly

**Dataset Code Comments:**

The authors are encouraged to release their full evaluation code, particularly for the proposed BMMR-Verifier framework. This would greatly benefit the community by enabling standardized comparisons and deeper analysis.

**Ethical Comments:**

NO.

**Ethical Considerations:**

No, there are no or only very minor ethics concerns

**Final Justification:**

My major concern is answered by the authors, and I keep my rating.

**Limitations Weaknesses:**

Some of the conclusions, such as the claim regarding “overthinking” in Phi-3.5-Vision-Instruct and QVQ-72B, lack rigorous justification.

The analysis would benefit from more quantitative evidence or controlled experiments to substantiate such observations.

**Strengths Contributions:**

1. This work presents a significantly larger multimodal reasoning dataset compared to existing multi-discipline datasets.
2. The authors conduct extensive evaluations with various LLMs and provide insightful findings, offering valuable guidance for future research in multimodal and multi-discipline reasoning research.

---

> ### Author Rebuttal · Authors · 2025-07-31
>
> Thanks for your valuable review, feedback, and your recognition of our work.
>
> ---
> > Q1: Question about the "overthinking" conclusion and hallucination.
>
> A1:
>
> - We sincerely appreciate your suggestion and feedback. In addition to qualitative evidence, we conducted a quantitative comparative analysis. Specifically, we statistically analyzed different percentiles and the average output length during inference for Phi-3.5-vision-Instruct, Phi-4-multimodal-Instruct, QVQ-72B-Preview, and Qwen2.5-VL-72B-Instruct. We also provided the evaluation performance of these models on the benchmark.
>
> | Model                          | 25% Output Length | 50% Output Length | 75% Output Length | AVG Length | AVG ACC |
> |--------------------------------|-------------------|-------------------|-------------------|------------|---------|
> | Phi-3.5-vision-Instruct     | 1124              | 1788              | 2790              | 4240.8     | 3.88    |
> | Phi-4-multimodal-Instruct   | 963               | 1496              | 2006              | 3795.75    | 12.82   |
> | QVQ-72B-Preview                | 3825.25           | 6120.5            | 10017.5           | 13043.17   | 23.31   |
> | Qwen2.5-VL-72B-Instruct     | 678               | 1185              | 1653.75           | 1216.1     | 38.22   |
>
> - From the experimental results, it can be observed that although Phi-3.5-vision-Instruct generates longer outputs than Phi-4-multimodal-Instruct, its performance is inferior. Similarly, QVQ-72B-Preview produces outputs that are **over ten times longer** (in average length) than those of Qwen2.5-VL-72B-Instruction, yet **its performance is worse**. Through in-depth case studies and observations, we found that QVQ-72B-Preview and Phi-3.5-vision-Instruction often generate redundant outputs. They may **even overturn their correct answers** during "overthinking" or "over-reflection", leading to performance degradation. Additionally, these models might "fabricate" extra conditions and content through hallucinations, which also contributes to performance decline. We also statistically analyzed error types in Section 6.2 of the paper.
>
> - Recently, relevant work has discussed this issue [1][2] and analyzed the impact of overthinking on reasoning efficiency. Regarding hallucinations, recent studies pointed out that hallucinations in multimodal models increase with longer reasoning processes [3]. Therefore, improving reasoning efficiency in multimodal setting to reduce both overthinking and hallucinations is a worthwhile research direction.
>
> - Following your suggestions, we will incorporate relevant qualitative comparative experiments and case studies into the paper to strengthen its validity. Thank you again for your feedback!
>
> ---
> > Q2: Question about the number of annotators in Line 246–247.
>
> A2:
>
> - We recruited 40 undergraduate and postgraduate students from diverse academic backgrounds and assigned them reasoning paths in different disciplines according to their majors for process evaluation. During the evaluation, we provided annotators with reference answers and reasoning paths for the corresponding questions to ensure the accuracy of the annotation. Thank you again for your suggestion. We will include this information in the manuscript.
>
> ---
> > Q3. Figure Readability.
>
> A3:
>
> -  We will refine the figure representation in the next revision. Thank you for your valuable feedback！
>
> ---
> > Q4. Accessibility
>
> A4:
>
> - Thank you for pointing out this. We will ensure that **all associated code and data are made publicly accessible**. We have prepared the code for BMMR-Verifier, which includes the training and inference scripts. Due to the constraints of the rebuttal, which prohibit attachments or supplementary links, we will release all materials publicly in the near future. We hope our work would provide a significant contribution to the research community.
>
> ---
>
> Thank you again for your feedback. Based on your suggestions, we have added additional experiments across different tasks to address your concerns. Please feel free to let us know if you have any further questions. If you find our revisions satisfactory, we kindly ask you to consider raising your score.
>
> ---
>
> [1] Do NOT Think That Much for 2+3=? On the Overthinking of o1-Like LLMs
>
> [2] Kimi k1.5: Scaling Reinforcement Learning with LLMs
>
> [3] More Thinking, Less Seeing? Assessing Amplified Hallucination in Multimodal Reasoning Models

---

### Official Review · Reviewer_qzyR · 2025-07-03

**Rating:** 4
**Confidence:** 3

**Summary:**

In this paper, the authors create a new dataset, a Bilingual Multimodal Multi-Discipline Reasoning dataset (BMMR), which supports the evaluation of multimodal foundation models in college-level, multidisciplinary knowledge, understanding, and reasoning. BMMR comprises 110k items spanning 300 UNESCO-defined subfields across 8 high-level disciplines and consists of training and test splits. To scale up the dataset, the authors prepared a specific verifier, BMMR-Verifier, to conduct human-in-the-loop for the dataset creation. Experimental results from 24 models spanning 12 series on BMMR indicate the difficulty of BMMR and its usefulness in evaluating strong models, as well as the importance of the training split in BMMR.

**Additional Feedback:**

How many instances are actually annotated by humans through the human-in-the-loop?

**Dataset Code Accessibility:**

Partly

**Dataset Code Comments:**

It's accessible at least. However, it lacks a detailed document.

**Ethical Considerations:**

No, there are no or only very minor ethics concerns

**Final Justification:**

Not all data instances are verified by humans. However, considering the entire dataset size, checking all instances is not so easy. Thus, I've updated the recommendation score.

**Limitations Weaknesses:**

- This is a severe issue where only partial data is annotated by humans in BMMR. Thus, it degrades reliability.

**Strengths Contributions:**

- The proposed dataset, BMMR, has an advantage in its scale compared to conventional datasets.
- To efficiently scale up the dataset size, the authors incorporate human-in-the-loop into the dataset creation.
- Since the dataset is based on UNESCO, the information included from the data source is reliable.
- The authors conducted a comprehensive evaluation by comparing 24 models spanning 12 series.
- Experimental results demonstrate the challenges of BMMR and highlight the value of utilizing this data for recent advanced models.
- The experimental results also demonstrate the importance of fine-tuning on the training data created by this work.

---

> ### Author Rebuttal · Authors · 2025-07-31
>
> Thank you very much for your valuable time and feedback. We are very pleased that you recognize the comprehensiveness, scalability, solidity of our work, as well as its value for advancing state-of-the-art models. In response to your questions, we would like to offer the following clarifications and responses:
>
> > Q1: Question about partial data in BMMR is annotated by humans.
>
> A1:
> - Thank you very much for pointing this out, and we apologize for not making it clear. We would like to offer the following clarification: Due to resource constraints, we were unable to conduct high-quality manual annotation at scale for the full 110k samples. Instead, we designed **a rigorous pipeline for data collection, processing, and quality control**. Our data is primarily sourced from both physical and digital media—such as books, exams, and quizzes—which often come with some degree of inherent annotation. We cleaned and processed these materials through a designed pipeline.
>
> - The main manual effort focused on quality controlling, and **we adopted a human-in-the-loop strategy** to reduce annotation costs with the help of large language models. Specifically, we employed a cascaded LLM filtering mechanism (see Appendix A) to retain a subset of examples that even the strongest models struggled with. These examples may include both noisy data and genuinely challenging, high-quality samples.
>
> - To annotate and verify these samples, we enlisted 40 annotators—30 undergraduate and 10 graduate students from diverse academic backgrounds. They were allowed to use any tools (e.g., search engines) and were instructed to follow a strict quality-first principle: only retaining samples they were highly confident in. Each annotated item underwent triple-blind cross-validation, and only samples with over 50% agreement were retained. This resulted in a curated set of difficult yet high-quality data.
> Although we could not manually check the entire dataset, our pipeline enabled us to curate a relatively high-quality dataset of **110k samples**. Similar large-scale dataset construction works [1][2] also do not fully rely on human annotation. Given that our primary goal is to contribute high-quality, multi-disciplinary data to the community, we believe our pipeline is a reasonable compromise under current constraints. We think we have made our best effort, and future users can build upon this foundation with additional processing based on their available resources and funding.
>
> - We will include these clarifications in the paper and continue working on further improvements. Thanks again for pointing it out!
>
> [1] MDBench: A Synthetic Multi-Document Reasoning Benchmark Generated with Knowledge Guidance
> [2] mCSQA: Multilingual Commonsense Reasoning Dataset with Unified Creation Strategy by Language Models and Humans
>
> ---
>
> > Q2：How many instances are actually annotated by humans through the human-in-the-loop?
>
> A2:
> - As mentioned in our response to Q1, the data we collected comes from both print and digital media such as books, exams, and quizzes, which inherently include annotations. Our manual effort was mainly focused on quality checking and control after applying **a cascaded model-based filtering process** based on model pass rates. Specifically, out of the final **110k samples**, **21.5k were manually checked**. Due to extensive rework, iterations, and redesigns of the pipeline—as well as the fact that our annotators were not working full-time—**the entire data processing effort took over four months**. Thank you very much for your question again! We will revise and supplement the manuscript accordingly to make the paper more complete.
>
> ---
>
> > Q3: Lack of detailed document.
>
> A3:
> - We anticipate our work will provide a significant contribution to the research community. To this end, we will ensure that **all associated code and data are made publicly accessible**.
> - We appreciate you highlighting the need for more detailed documentation for our existing code and datasets to facilitate easier community evaluation. In response, **we have prepared comprehensive documentation** that elaborates on the following: 1) how to generate answers with the model on our dataset, 2) how to extract the final answers from the model's output, and 3) how to perform final judgments based on the extracted answers and ground truth.
> - Due to the constraints of the rebuttal, which prohibit attachments or supplementary links, we will release all materials publicly in the near future. Furthermore, to enhance usability, **we have integrated our dataset and evaluation code into the well-documented and influential open-source multimodal evaluation framework**, open-compass/VLMEvalKit. This integration will provide a more unified interface and allow the community to comprehensively evaluate the reasoning capabilities of their large multimodal models in a familiar manner.
>
>
> Thank you again for your feedback. Based on your suggestions, we have made every effort to clarify the relevant points to address your concerns. Please feel free to let us know if you have any further questions. If you find our revisions satisfactory, we kindly ask you to consider raising your score.

---

> > ### Author Response · Authors · 2025-08-05
> > **Friendly Reminder about the Discussion**
> >
> > Dear Reviewer qzyR,
> >
> > Thank you for your valuable time and great efforts in reviewing our work and for your insightful questions. As the deadline for the reviewer–author discussion period approaches, we have not yet received your response to our rebuttal. We hope that our answers have helped to clarify the points raised, and we look forward to hearing from you. Please let us know if you require any further information or have any additional concerns, and we will try our best to address them. Thank you again!
> >
> > Best regards,
> >
> > Authors of Submission 2221

---

> ### Author Response · Authors · 2025-08-06
> **Kind Invitation to Discussion**
>
> Dear Reviewer qzyR,
>
> Thank you again for your thoughtful and constructive reviews. We appreciate the time and effort you’ve dedicated to evaluating our work.
>
> May we kindly ask whether our rebuttal has adequately addressed your concerns? If there are any remaining questions or points you'd like us to clarify, we’d be happy to provide additional information or make further improvements.
>
> Best regards,
>
> Authors of Submission 2221

---

> ### Comment · Reviewer_qzyR · 2025-08-06
> **I understand the actual annotation steps**
>
> Thank you for the detailed explanation of the human annotation steps. As I thought, not all data instances are checked by humans. However, one-fifth of the data is verified by humans. This ratio is not so small considering the entire dataset size. Based on this fact, I've decided to update my score.

---

> > ### Author Response · Authors · 2025-08-06
> > **Thanks for Reviewer qzyR's positive response and recognition!**
> >
> > We are deeply grateful for your response. Your positive feedback has provided us with significant encouragement! We will further offer specific explanations regarding our manual annotation methods in the final version.

---

### Note · Authors · 2025-08-16

We sincerely appreciate the thoughtful feedback from all reviewers and the opportunity to address their concerns, which have significantly strengthened our work. Here, we summarize key points from the discussions to provide closure:

First, regarding **dataset quality and annotation reliability**—a primary concern—we clarified that while the full 110k BMMR samples were not all manually annotated, we implemented a rigorous pipeline: 21.5k critical samples (after cascaded LLM filtering) underwent triple-blind cross-validation by 40 annotators (30 undergraduates, 10 graduates) with diverse academic backgrounds, retaining only those with >50% agreement. This human-in-the-loop strategy, combined with sourcing from trusted educational materials (books, exams), balances scale and quality, aligning with practices in large-scale dataset construction.

Second, **addressing the "overthinking" conclusion**, we supplemented quantitative evidence showing models like Phi-3.5-Vision-Instruct and QVQ-72B generate excessively long outputs (avg. 4.2k and 13k tokens) with lower accuracy (3.88% and 23.31%) compared to more concise models like Qwen2.5-VL-72B (1.2k tokens, 38.22% accuracy). Case studies and error analysis confirmed redundant reasoning and hallucinations as contributing factors, supported by recent paper on overthinking and multimodal hallucinations.

Third, on **generalization of fine-tuning** we conducted experiments mixing BMMR with MMPR, showing InternVL2.5-4B improved by 1.78% (BMMR alone) and 3.52% (BMMR+MMPR) on MMMU and MathVerse, demonstrating transferable multi-disciplinary knowledge. This addresses overfitting concerns and validates the training set’s value.

Regarding accessibility, we commit to releasing full code (including BMMR-Verifier), detailed documentation, and integrating with VLMEvalKit. We’ve clarified 40 annotators were involved and will improve figure readability as suggested.


We believe these revisions address all key concerns, and we’re committed to supporting the community with accessible resources. Thank you again for your guidance—we hope BMMR will contribute meaningfully to multimodal reasoning research.

---

### Decision · Program_Chairs · 2025-09-18

**Decision:**

Accept (poster)

**Comment:**

The paper introduces BMMR, a new large-scale, bilingual dataset designed for evaluating the reasoning capabilities of large multimodal models. BMMR could be a valuable resource considering its scale and diversity. The paper presents extensive experiments conducted on different models, indicating significant room for improvement when tested on the proposed dataset.